# Usefulness of Hulled Wheats Grown in Polish Environment for Wholegrain Pasta-Making

**DOI:** 10.3390/foods10020458

**Published:** 2021-02-19

**Authors:** Aneta Bobryk-Mamczarz, Anna Kiełtyka-Dadasiewicz, Leszek Rachoń

**Affiliations:** 1PZZ LUBELLA GMW Sp. z o.o. Sp. k., ul. Wrotkowska 1, 20-469 Lublin, Poland; A.Bobryk-Mamczarz@maspex.com; 2Department of Plant Production Technology and Commodity Science, University of Life Sciences in Lublin, Akademicka 15, 20-950 Lublin, Poland; leszek.rachon@up.lublin.pl

**Keywords:** innovative pasta raw materials, durum, emmer, spelt, pasta

## Abstract

The best pasta raw material is durum wheat (*Triticum turgidum* subsp. *durum* (Desf.) Husn.). Recently, old wheat species have also attracted interest. The aim of the study was to evaluate their usefulness for industrial pasta production. The technological characteristics of grains and the organoleptic characteristics of pasta obtained from hulled emmer (*T. turgidum* subsp. *dicoccum*) and spelt (*T. aestivum* ssp. *spelta*) were determined and compared to durum wheat, as a standard pasta raw material, and common wheat (*T. aestivum*). All wheats were grown under identical conditions. The hardness of kernels was assessed using the practical size index, wheat hardness index, torque moment, milling work of 50 g of flour, semolina yield, and starch damage. The technological and nutritional values of semolina, i.e., protein and ash content, wet gluten yield and quality, and falling number, were determined. Moreover, the organoleptic characteristics of cooked pasta were analysed in terms of appearance, colour, taste, smell, and consistency. The milling parameters of emmer were comparable to those of durum wheat; moreover, the content of protein, gluten, and ash was higher in emmer. Spelt was found to be similar to common wheat. Hulled wheats, especially emmer, show good quality parameters and can be an alternative raw material for industrial pasta production.

## 1. Introduction

It is widely believed that the best raw material for pasta production is durum wheat (*Triticum durum* Desf.) due to the hardness and vitreousness of its grains, quality of gluten proteins, high yellow pigment content, as well as its light and thin bran layer [1,2]. In recent years, however, much attention has been given to the old species, i.e., hulled wheats such as spelt, emmer, or einkorn, to make traditional products. Hulled wheats are characterised by a lower yield but many of them have a high nutritional value, including higher protein content and wet gluten yield as well as higher content of macro-, micronutrients, and vitamins [3,4,5,6,7,8]. Regardless of the species, wheat kernels and the resulting semolina should meet certain requirements of pasta producers. The content and quality of protein and gluten, kernel vitreousness and hardness, yellow pigment content, and falling number are relevant. During the production of pasta, the granulation of semolina, its ash content, and the degree of starch damage are also of importance [9]. Pasta producers from different countries have similar requirements for semolina dedicated to the production of pasta. Polish manufacturers of pasta often use the quality guidelines for wheat grain and semolina contained in the non-obligatory Polish Standards [10,11], where, in the case of semolina, recommendations are total ash content max. 0.9%; wet gluten yield min. 30% and its deliquescence up to 13 mm; falling number, being a measure of the amylolytic activity of durum wheat grain, at a minimum level of 300 s; and bulk density of grain at least 75 kg hl^−1^. Italian law indicates protein content as min. 10.5% and max. 0.9% ash content [12]. On the other hand, Sieber [13], in turn, states that pasta producers in Germany require durum wheat to contain more than 14% protein, grain vitreousness above 75%, parameter b* above 22, and falling number above 220 s.

According to the legislation of France, Greece, and Italy, pasta for local markets can be produced only from durum wheat, without any admixtures of common wheat. In countries such as the USA, Canada, Australia, or Spain, only durum wheat is used by choice in the production of pasta [1,7,14]. In other European countries, pasta producers also use other available raw materials, including high-quality cultivars of common wheat. Other types of wheat may also potentially be the pasta raw material [15,16], but insufficient data are currently available on the use of hulled wheat as a substitute for durum and common wheat in industrial pasta making and their quality [6].

Durum wheat has specific climatic requirements. It grows best in dry, hot, continental climates; therefore, it is mainly grown in the Mediterranean basin, North America, and Kazakhstan. For several years now, it has also been successfully cultivated in Central Europe. Attempts have also been made to obtain durum wheat cultivars for cultivation under temperate conditions [17,18,19,20]. The choice of raw materials for the pasta industry is associated not only with their quality, but also their supply. This can be problematic in cases of less yielding and less widespread hulled wheats; climatic constraints on durum cultivation should also be considered [21]. The cultivation of common wheat (*Triticum aestivum* ssp. *vulgare* L.) is the most widespread in the world [22]. Some cultivars of common wheat are of high technological value and are used not only as a bakery raw material, but also for pasta production in some parts of Europe and the world [22,23,24,25]. The aim of the present study was to determine the usefulness of two species of hulled wheats, emmer and spelt, for the pasta industry, as compared to two cultivars of durum wheat grown in Poland under temperate conditions, with the indication that 'SMH87' is a local cultivar. [26].

## 2. Material and Methods

### 2.1. Plant Material

In order to assess the suitability of the hulled wheats (spelt and emmer) for pasta production, the technological characteristics of kernels were determined and compared with durum wheat, considered the best pasta raw material and common wheat, and now also widely used in the pasta industry (Table 1).

To eliminate the effects of the changeable environmental conditions of raw material production on the quality and usability of the kernels, a field experiment was conducted, in which the same agrotechnical, soil, and climatic conditions were maintained for each cultivar tested, as previously described [21]. The experiment was continued for three consecutive years (2015–2017) in the locality of Hopkie (50°30′28″ N 23°39′40″ E; 221 m a.s.l., Lubelskie Province, Poland) on rendzina soil. The experiment was set up using a random block design, with three replicates, on plots with area of 0.17 ha. The conducted soil tillage was typical of a conventional tillage system and the fertilisation and cultivation of the plantations were used as modern agriculture and managed in accordance with Good Agriculture Practice. The suitability of kernels of individual cultivars for pasta production was assessed annually. After harvesting, the kernels were cleaned and dried to a constant humidity of 13%.

### 2.2. Kernel Hardness Characteristics

The milling value of kernels of each cultivar was evaluated using the following hardness characteristics:The particle size index (PSI) expressed as % of the flour produced under the standard grain milling conditions obtained using a Quadrumat Junior mill (Duisburg, Germany). The grain was ground in a mill with grinding gaps of I–II at 0.8 mm; II–III at 0.3 mm; III–IV at 0.1 mm and roll grooves of I and II at 5R/cm; III and IV at 8 R/cm. The obtained grist was sieved in a laboratory sifter on a sieve with a mesh size of 500 μm, so that we received wheat bran and unpurified semolina. Purified semolina (extract 50% ± 2% in relation to the grain) was obtained on the principle of self-sorting and was undersown on sieves wrapped with gauze at 0.8, 0.5, and 0.35 mm. Higher PSI values correspond to the grains of lower hardness.Using a Brabender hardness tester (Nossen, Germany) determined:The torque value expressed as the maximum height of the graph in Brabender units (BU).The milling work required for fragmentation of 50 g of the grain sample, read as a function of the surface plotted by the recorder.The amount of flour produced with a particle size of <120 μm on the laboratory sifter (%).The wheat hardness index (WHI) expressed as the ratio of torque in BUs to the quantity of flour (%),The yield of non-purified semolina (%) using a Quadrumat Junior mill (Duisburg, Germany).

### 2.3. Quality of Wheat Semolina

From the wholegrain samples, semolina was separated reaching—for all of the tested wheat species—a constant yield of 50% in relation to the grain. The usefulness of semolina was evaluated by determining:The total protein content (%) using the Kjeldahl method according to PN-EN ISO 20483 [27].The yield of wet gluten (%), its elasticity, and deliquescence according to PN-77/A-74041 [28]. To 50 g of semolina, 25 cm^3^ of tap water at 20 °C was added and the dough was kneaded. The dough was rolled into a ball by hand and placed in a steamer for 20 min. After this time, the dough ball was kneaded under running water until the starch was completely washed out (until the water showed no reaction to the presence of starch with Lugol’s solution). The obtained gluten ball was pressed by hand to remove excess water, and its weight was determined on a laboratory balance with an accuracy of 0.01 g. The gluten content was converted into 100g of semolina. In order to determine the elasticity, 5 g of the washed gluten was weighed with an accuracy of 0.01 g and formed into a 2 cm long roll. The roller was taken in two hands with the tips of the fingers and brought closer to the millimetre scale so that the lower end of the roller fell to the zero point of the scale in the upper part of the measure. Then, with the fingers of one hand, it was pulled down slightly to the 5 cm point, then the lower end of the roller was released and the behaviour of the pulled-out gluten roller was observed. Gluten elasticity is defined in degrees: 1st degree—elastic gluten, showing the ability to stretch up to 5 cm and return to the zero point of the scale; 2nd degree—moderately elastic gluten, showing the ability to stretch up to 5 cm and return only to half the length, i.e., up to 2.5 cm; 3rd degree—inelastic gluten, showing the ability to stretch, but not shrinking completely, sagging and showing the ability to stretch further; 4th degree—inelastic (short) gluten, breaks before stretching up to a length of 5 cm. Gluten deliquescence was determined as follows: 5 g of gluten was balled and placed on a glass plate with a millimetre mark underneath. The ball diameter was measured in two perpendicular directions. The plate was covered with a glass beaker and placed in an oven at 30 ° C for 60 min. After this time, the ball diameter was measured again. Gluten deliquescence is expressed in mm as the difference between the final and initial ball diameters.The falling number applying the Hagberg–Perten method according to PN-ISO 3093 [29].The degree of starch damage (%) using SD Matic (Chopin, Villeneuve-la-Garenne, France), according to AACC 76-31 [30].The total ash content (%) according to PN-ISO 2171 [31].Colour using a CR-410 Chroma Meter (Konica-Minolta, Tokyo, Japan) in the CIE L*a*b* system, where L* is a measure of lightness (ranging from 0 for ideal black to 100 for ideal white); a*, where negative values indicate green and positive values indicate red; and b*, where positive values indicate yellow. While interpreting the numerical values in the CIE L*a*b* system, it should be assumed that the higher the b* value, the more yellow the sample and the higher the L* value, the lighter the sample. In the paper, we presented only the values of b*, reflecting the yellow colour of the sample, in correlation with L*, responsible for the lightness of the sample. The third element of chromaticity in the CIE L*a*b* system, i.e., a*, which determines the intensity of the red colour, was neglected as its values were around zero (0.13–1.95) and the parameter itself is less important for the quality of pasta.

### 2.4. Preparation and Organoleptic Evaluation of Pasta

The wholegrain pastas were prepared under repeatable laboratory conditions from each of the wheat cultivars tested to determine their organoleptic characteristics. The following procedures were observed: 80 mL of water at room temperature was slowly added to 200 g of wholegrain wheat semolina obtained after the one-step 20-s milling (Thermomix Vorverk, Wollerau, Switzerland), thus a characteristic pasta dough in the form of a crumble with a moisture content of about 38% (deficient in water) was obtained. The pasta dough was formed in a steel kneading-trough using a rotating agitator for 3 min, embossed through a matrix, and formed into a rotini shape, which was then cut with a slidable knife into four-centimetre pieces (Figure 1). The products were pre-dried at 35 °C for 30 min and then dried in a food dryer at 60 °C (±2 °C) for 6 h to a standard humidity of 12% (Figure 2).

The pasta of each sample was cooked separately in slightly salted water (14 g of NaCl for 2 l of water according to PN-93/A-74130 [32]) over the prescribed minimum cooking time (5 min.), after which the cooked pasta is *al dente* and ready to eat. The organoleptic characteristics were evaluated by a team of five professional certified sensory experts with confirmed sensory sensitivity, professionally involved in organoleptic analysis. The 100-g samples of pasta obtained from each of the cultivars tested were assessed in terms of appearance, colour, taste, smell, and consistency in accordance with PN-87/A-74131 [33]. Each feature was scored on a scale of 1 to 5. The arithmetic means of five evaluations gave the final result for the pasta of each wheat.

### 2.5. Statistical Analysis

The results were statistically analysed using the analysis of variance (ANOVA) and Statistica 12 PL software, assessing with Tukey’s post hoc HSD (honest significant difference) test; *p* ≤ 0.05 was considered statistically significant.

## 3. Results and Discussion

### 3.1. Grain Milling Value

The technological suitability of wheat for pasta production can be assessed based on the milling value of grains and the physicochemical parameters of semolina. The highest torque value, corresponding to the wheat of harder grains, was found for the emmer wheat (372 BU) followed by 'Floradur' (358 BU) and 'SMH87' (345 BU) (Table 2). The biggest work (1201 J) required for fragmentation of 50 g of a grain sample was found for 'Floradur', followed by 'SMH87'—1155 J; the results differed significantly. The work needed for fragmentation of the emmer wheat was 9.6% lower, as compared to 'Floradur'. In the case of soft endosperm wheats, i.e., spelt and common wheats, both parameters mentioned above were lower, thus more fine flour < 120 µm was produced during milling. Both wheat cultivars tested (durum and emmer) were also characterised by higher yields of wholegrain semolina, as compared to spelt and common wheat. The above results were consistently confirmed by the WHI and PSI. The highest WHI was recorded for durum wheat cultivars: 'Floradur'—160; 'SMH87'—143; emmer—131. The PSI, which expresses the percentage of flour produced during fragmentation, was the lowest one for wheats with harder endosperms. According to Rachoń [34], the grain of higher hardness wheats is best for pasta production, due to its higher ability to form semolina, the granulation and roughness of milling products, and the amount of fine flour produced during milling. The author has reported a higher torque value, lower amounts of fine flour, a higher WHI, and a lower PSI for hard durum wheat cultivars and lines compared to common wheat 'Sigma'. Moreover, Cacak-Pietrzak and Gondek [35] as well as Wójtowicz et al. [24] have demonstrated a higher hardness of common wheat grains, as compared to spelt, which is consistent with our results.

Furthermore, the degree of starch damage in semolina depends on the hardness of kernels. According to Dziki et al. [36], the flour obtained from harder grains is characterised by a higher degree of starch damage, as compared to the soft wheat flour, which is confirmed by our results (Figure 3). For both sifting granulations of purified semolina (through a sieve of 95 and 180 μm), the highest degree of starch damage was found in 'Floradur' and 'SMH87', followed by emmer (8.0 and 5.8%, 7.9% and 5.6%, and 7.1 and 4.4%, respectively). Lower starch damage was observed in the common wheat and the lowest one in the spelt (6.0 and 4.1% and 4.0 and 2.6%, respectively).

Milling conditions have a significantly greater impact on the degree of starch damage than the choice of wheat cultivars [37,38]. The degree of starch damage is a determinant of milling quality assessment, which may cause the water absorption of semolina during kneading the dough to be too high, and during pasta cooking, too much water-soluble amylose is released into the solution. According to Szafrańska [39], the damaged starch absorbs more water during dough formation, as compared to the undamaged starch; hence, less water is left for proper gluten network development. During the milling of durum wheat into semolina for the production of pasta, the aim is to minimise starch damage. The optimal degree of starch damage also depends on the amount of total protein [39]. Therefore, in the case of wheat milling for pasta production, the degree of starch damage should be as low as possible. Our results indicated that the degree of starch damage increased with an increase in granulation for all the samples tested. The highest degree of starch damage (in the material of the same granulation) was recorded in the case of both durum and emmer wheat cultivars, and the lowest in spelt. Much lower starch damage in products of the same species and cultivars, but with higher granulation, recommends such grinding for pasta purposes, which results in as little flour as possible and as much grist as possible.

### 3.2. Quality Parameters of Semolina

#### 3.2.1. Protein Content and Gluten Yield

The protein content in the raw material is one of the most important parameters determining its suitability for pasta production [1,34,40]. The highest total protein content in purified semolina was found for emmer—18.0%; significantly lower values were determined in spelt semolina—14.8%; and in both durum wheat cultivars—'SMH87' and 'Floradur'—4.1 and 4.3 percentage points (p.p.), respectively, as compared to emmer (Table 3). The lowest total protein content was found in common wheat semolina—12.2%. The highest amount of gluten was washed out from hulled wheat semolina: in emmer, the percentage of gluten was 38.0%, a significantly lower amount (by 4.6 p.p.) was found in spelt. The amount of gluten washed out from wheats of both durum cultivars was comparable (28.7–28.8%); the lowest yield of gluten was observed for common wheat—24.3%, which was lower by 13.7 p.p. than the yield observed in emmer. Likewise, in the studies by Branković et al. [41], Woźniak [42], Geisslitz et al. [43], and Rachoń [20], the durum wheat contained more total protein and gluten than the common wheat. Majewska et al. [44] reported a higher content of wet gluten in flours from seven spelt cultivars and a higher total protein content (except for one cultivar—Celario), as compared to common wheat flour. In contrast, Sobczyk et al. [45] reported a lower content of protein and gluten proteins in spelt than in common wheat; in turn, Frakolaki [46] reports that spelt had more protein but less gluten than common wheat. According to Suchowilska et al. [47], the total protein content was higher in emmer than in spelt, which was confirmed in our study.

#### 3.2.2. Quality of Gluten

The quality of gluten was assessed by determining the deliquescence and elasticity of gluten. The lowest deliquescence was determined in gluten from the common wheat semolina—2.8 mm; its value was significantly different compared to other species. The deliquescence of gluten from 'Floradur' (4.5 mm) was not significantly different from the value observed for spelt semolina—4.8 mm (Table 3). Furthermore, the deliquescence of gluten from spelt semolina was not significantly different from that observed in 'SMH87'—5.6 mm. The highest deliquescence was found in emmer gluten—13.0 mm. Emmer gluten was characterised by the highest elasticity (III degree). The remaining species were characterised by second-degree elasticity. Rachoń [34] observed deliquescence of 7–13 mm in eight lines and cultivars of durum wheat. The author has emphasised that gluten in the pasta industry cannot be too short and strong (deliquescence should not be too low) or too weak (deliquescence should not be too high as well). According to the study results reported by Rachoń et al. [2], the durum wheat flour was characterised by gluten deliquescence of 6.3–6.6 mm; in the common wheat flour, this value was 1.5 mm while in the spelt flour—4.0–4.4 mm.

#### 3.2.3. Falling Number

The highest falling number was recorded for 'Floradur'—506 s. The falling number for 'SMH87' was significantly lower—477 s. The falling number of semolina obtained from emmer was 452 s, and from spelt—388 s. The lowest falling number was found in common wheat semolina—375 p. The Hagberg falling number indicates the activity of alpha-amylase in the kernel; in the case of sprouted grain, the falling number is expected to be lower than in grains with low alpha-amylase activity [48,49]. According to Woźniak [42], the falling number in durum wheat was higher than 300 s, regardless of the level of agrotechnics. In the study by Sobczyk et al. [45], the falling number in spelt flour was 257–364 s, while in common wheat flour—271 s. Moreover, Majewska et al. [44] have reported a falling number of 215–315 s in spelt flour and of 296 s in ordinary wheat flour. According to Krawczyk et al. [50], the value of this parameter was 270–331 s in spelt flour and 296 s in common wheat flour. Stolickova and Konvalina [51] in their studies on organic farms in the Czech Republic found the lowest falling number in the control sample, which was common wheat—245 s; in emmer, this number was 238–338, and in spelt— 304–356 s. Higher values for falling number in the case of refined semolina in relation to the whole grain may indicate that debranning and germ removal during the milling process reduce the activity of amylolytic enzymes and thus, increase the falling number. Obuchowski [52] states that semolina of the quality indicated for pasta making should have a falling number of 350–450 s. Polish standards [10,11] indicate the value of the falling number in the raw material for pasta production at a minimum of 250 s for common wheat and a minimum of 300 s for durum wheat. Sjoberg et al. [53] distinguish low-quality wheat genotypes as this with falling number below 300 s. The low falling number, in addition to the risk of excessive darkening of the pasta, may affect its stickiness and the formation of clumps due to the process of excessive starch degradation [52]. In our research, the falling number levels in all semolinas met the assumptions for production of over 300 s as recommended above. It can therefore be concluded that all cultivars of the species compared met the requirements for the production of pasta in terms of this parameter.

#### 3.2.4. Ash Content

Statistical analysis showed that the highest total ash content, i.e., residues from burning a sample of grain or flour containing minerals, was observed in emmer semolina—1.27%. A lower score was obtained for 'SMH87'—0.85%; and 'Floradur'—0.80%. Semolina from wheats with soft endosperms was characterised by the lowest mineral content. Similarly, the lowest ash content in common wheat compared to durum, emmer, and spelt was obtained by Geisslitz et al. [43]. The content of ash in spelt wheat semolina was 0.70% and in common wheat semolina—0.63%. Likewise, according to Rachoń [34], the mineral content in durum wheat semolina was higher, as compared to common wheat. The author has indicated that due to the differences in the distribution of mineral compounds throughout the grain (higher content in durum endosperm compared to common wheat), durum wheat semolina contains more ash, compared to wheat flour, which is confirmed by earlier studies [21]. Even though the emmer wheat was also shown to be the richest source of minerals, there was less ash in the entire durum wheat grains than in spelt and ordinary wheat [8,21].

#### 3.2.5. Semolina Colour

The colour of semolina obtained from the wheats tested was found to be characteristic of individual wheat cultivars, while the differences in individual years of the study were slight (Figure 4). The semolina from both durum wheat cultivars studied was characterised by the highest value of b*, which describes the intensity of yellow colour desirable for pasta raw materials. This value fluctuated slightly over the years—the most yellow durum semolina was obtained in 2015. Podolska and Wyzińska [54] have also stressed the differences in the beta-carotene content of durum wheat in the individual years. Furthermore, Fu et al. [55] have observed higher values of b* (27.8–32.7) in Canadian West Amber durum (CWAD) semolina; the parameter depended on several factors, including the granulation of semolina—the finer the semolina, the lower its value at a given pigment content in the raw material. In contrast, Sieber et al. [56], in their study of 46 durum wheat lines collected in Germany, determined lower b* values (in the range 15.0–19.1). Likewise, Subira et al. [9] found b* in the range of 12.9–14.5 for wholegrain durum wheat flours grown under Italian and Spanish conditions, with higher values in modern than in old cultivars. Moreover, comparing the carotenoid content of the three wheat species, Piergiovanni et al. [57] demonstrated the highest carotenoid content in the durum wheat, followed by spelt; the lowest content was observed in emmer. According to Rachoń [34], the yellow pigment content in durum wheat cultivars and lines was 27.7% higher, as compared to the common wheat.

In our study, the lowest L*, determining the lightness of the samples, was observed in durum semolina in 2015, which means that these samples were slightly darker than those of 2016–2017. The semolina obtained from the emmer wheat was slightly darker, as compared to the durum semolina (lower L*), while the common wheat and spelt semolina was slightly lighter (higher L*). Fu et al. [55] have reported L* values between 83.8 and 85.5 for the CWAD semolina, i.e., darker than the semolina in our study. Fuad and Prabhasankar [58] have reported the highest L* value (the highest lightness) for common wheat semolina (85.8), followed by durum semolina (81.4); emmer semolina was the darkest one (74.2), which is consistent with our findings. The authors have pointed out that the highest value of L*, i.e., the lightest colour, of the common wheat semolina may be associated with the lowest bran fraction content and lower ash content determined for this wheat species.

Analysis of the b* and L* values of the semolina leads us to expect that the darkest pasta with a shade of yellow will be obtained from emmer wheat. The common wheat and spelt pasta should be expected to be light in colour, although of a low yellow intensity; the durum wheat semolina should provide the most optimal colour (in terms of colour and lightness), regardless of weather conditions during grain maturation.

### 3.3. Organoleptic Evaluation of Pasta

Pasta obtained from different wheat species differed in sensory characteristics, such as appearance, colour, taste, smell, and consistency (Figure 5). The highest average score (4.4 points) was found for wholegrain pasta from durum wheat 'SMH87': four out of five sensory experts rated it the highest compared to the other pastas. Slightly lower scores were reported for 'Floradur' and spelt pasta—4.2 and 4.1, respectively. Common wheat pasta received the lowest scores in terms of consistency, colour, appearance, and taste.

## 4. Conclusions

The results regarding grain hardness, semolina technological parameters, and organoleptic evaluation of pasta do not eliminate hulled wheats as a pasta raw material. Hulled wheats are not as good as durum wheat in any quality area, but many parameters, including the entire organoleptic characteristics of pasta, were better than in the common wheat, which is also a proper pasta raw material in some regions of the world. In many features, the differences between emmer and spelt proved important. Emmer had the highest protein, gluten, and ash content of all the cultivars studied. The yield of semolina, kernel hardness, the resulting milling parameters, and the degree of starch damage of the emmer wheat were closest to those of the durum wheat. However, the colour of the semolina and the organoleptic characteristics of the pasta were still weaker than those of durum. In the case of spelt, the parameters in question were similar to those of the common wheat.

In conclusion, the hulled wheats discussed, especially emmer but also spelt, may be considered alternative raw materials for industrial pasta production, provided that their supply is adequate. They are characterised by good quality parameters in the areas studied. These are interesting preliminary results which encourage deepening of the study with a larger number of cultivars and with larger amounts of grain in order to perform trials at the industrial level and also in mixtures with durum wheat.

## Figures and Tables

**Figure 1 foods-10-00458-f001:**
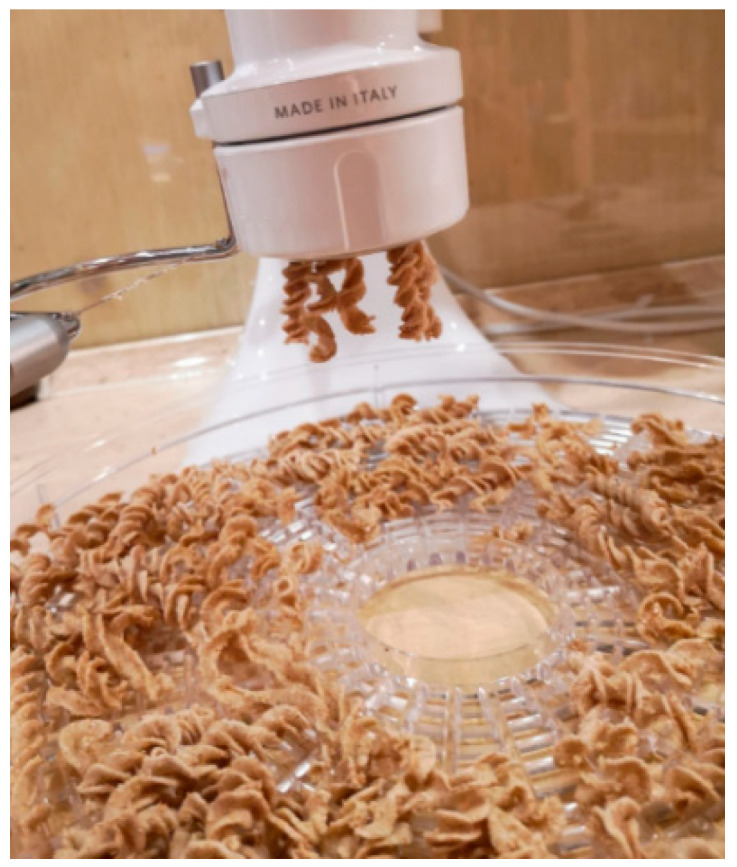
Pasta preparation (photo by A. Bobryk-Mamczarz).

**Figure 2 foods-10-00458-f002:**
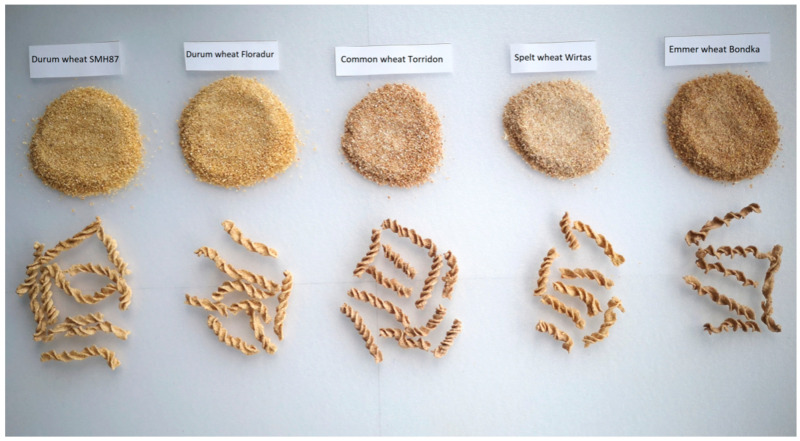
Wholegrain semolina and pasta of individual cultivars (photo by A. Bobryk-Mamczarz).

**Figure 3 foods-10-00458-f003:**
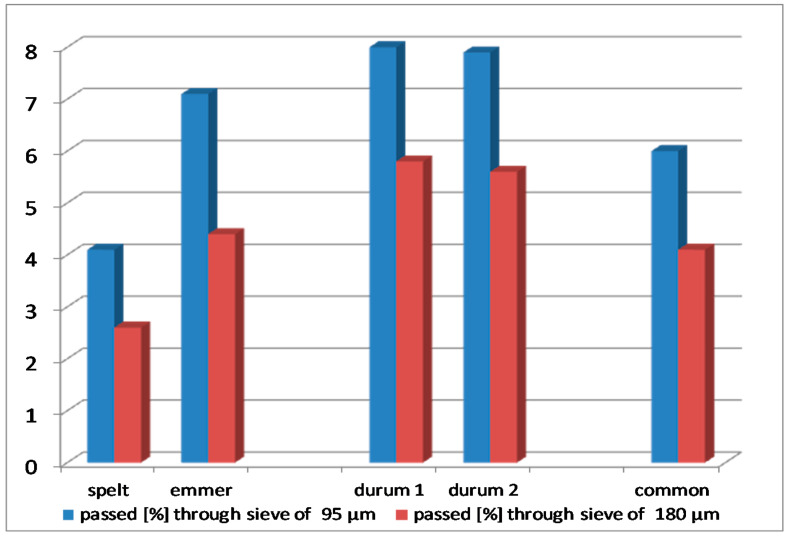
Starch damage degree.

**Figure 4 foods-10-00458-f004:**
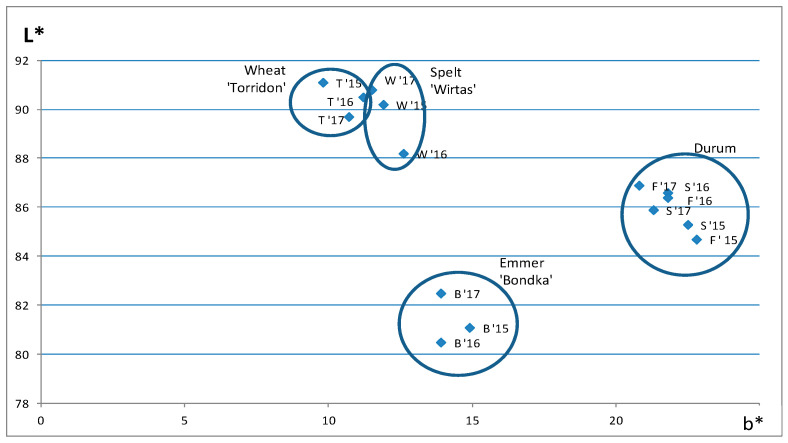
Values of L* responsible for sample lightness and of b* corresponding to yellow colour of purified semolina obtained from the wheats in the individual years of the study (2015–2017) defined in the CIE L*a*b* system. Explanations: W—spelt 'Wirtas'; B—emmer 'Bondka'; S—durum 'SMH87'; F—durum 'Floradur'; T—common wheat 'Torridon'; ‘15—2015 year; ‘16—2016 year; ‘17—2017 year.

**Figure 5 foods-10-00458-f005:**
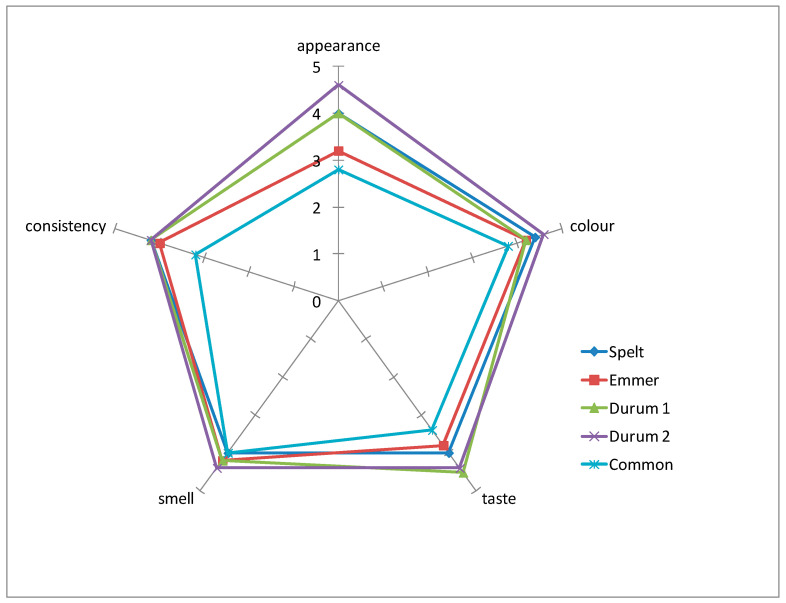
Organoleptic evaluation of pasta produced from individual wheat species.

**Table 1 foods-10-00458-t001:** Wheat species and cultivars tested.

Wheat Species	Botanical Latin Name	Cultivar
spelt	*Triticum aestivum* ssp. *spelta* (L.) Thell.	'Wirtas'
emmer	*Triticum**turgidum* subsp. *dicoccum* (Schrank ex. Schübl.) Thell.	'Bondka'
durum	*Triticum turgidum* subsp. *durum* (Desf.) Husn.	1. 'Floradur' 2. 'SMH87'
common	*Triticum aestivum* L.	'Torridon'

**Table 2 foods-10-00458-t002:** Kernel hardness characteristics (means of the years 2015–2017).

Wheat Species	Torque Value [BU]	Milling Work of 50 g of Flour [J]	Amount of Flour with a Particle Size <120 µm [%]	wholegrain Semolina Yield [%]	WHI	PSI
spelt	252 ^E^*	759 ^E^*	5.64 ^B^*	73.4 ^D^*	49 ^D^*	16.3 ^A^*
emmer	372 ^A^	1086 ^C^	2.89 ^C^	76.7 ^A^	131 ^C^	7.2 ^C^
durum 1	358 ^B^	1201 ^A^	2.34 ^D^	76.4 ^B^	160 ^A^	7.2 ^C^
durum 2	345 ^C^	1155 ^B^	2.38 ^D^	76.5 ^AB^	143 ^B^	7.0 ^C^
common	335 ^D^	966 ^D^	7.04 ^A^	75.0 ^C^	50 ^D^	12.9 ^B^

* Values denoted with the same letter are not statistically significantly different (*p* ≤ 0.05).

**Table 3 foods-10-00458-t003:** Quality parameters of fine semolina (means of the years 2015–1017).

Wheat Species	Protein Content [%]	Wet Gluten Yield [%]	Deliquescence of Gluten [mm]	Elasticity of Gluten [Degrees]	Falling Number [s]	Total Ash Content [%]
spelt	14.8 ^B^*	33.4 ^B^*	4.8 ^BC^*	II	388 ^D^*	0.70 ^D^*
emmer	18.0 ^A^	38.0 ^A^	13.0 ^A^	III	452 ^C^	1.27 ^A^
durum 1	13.7 ^C^	28.7 ^C^	4.5 ^C^	II	506 ^A^	0.80 ^C^
durum 2	13.9 ^C^	28.8 ^C^	5.6 ^B^	II	477 ^B^	0.85 ^B^
common	12.2 ^D^	24.3 ^D^	2.8 ^D^	II	375 ^E^	0.63 ^E^

* Values denoted with the same letter are not statistically significantly different (*p* ≤ 0.05).

## Data Availability

The data presented in this study are available on request from the corresponding author.

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
