# Peer review of "Usefulness of Hulled Wheats Grown in Polish Environment for Wholegrain Pasta-Making"

_foods, 2021, doi:10.3390/foods10020458_

Round 1

Reviewer 1 Report

The authors carried out a study to analyze the pasta-making quality of few hulled wheat genotypes. The paper needs significant changes and to be improved in several aspects, including using a higher number of genotypes, more appropriate methodologies, and better discussion of the data.  

Lanes 29-32: change this paragraph. It seems that the authors are saying that hulled wheats have received lot of attention for pasta production in the last year. That is not true. Hulled wheats have received lot of attention to make traditional products, but not real interest by the pasta-making industry.

Lanes 33-34: please be more objective. SOME hulled wheat genotypes have higher concentration of some components related with nutritional quality but not all. There are some naked wheats (landraces and modern wheat) with better nutritional quality than some hulled wheats. Please, do not over-generalize the results of some studies. Hulled wheats are good, but they are not better for everything, including grain quality.

Lanes 46-47: I cannot understand this well. Please, write it again.

Lane 50: is Floradur a “local” variety? I think this cultivar was developed in Austria.

Lane 50: are local cultivars

Lanes 29-50: the introduction is probably too short. The background given is not enough to understand why it is interesting to explore the pasta-making properties of hulled wheats. It is important to explain better why these lower yield and more expensive crops could be interesting to be used instead of durum wheat (lower price and better processing and end-use quality) by the pasta-making industry.

Lane 62: although the field trial is described in more detail in a previous paper, give some basic information of the trial in the current paper. Was it replicated? Type of design. What type of field management was applied?

Lanes 78-80: this is not clear. I do not understand well what type of sample was used for the analysis showed below. Particularly, I do not understand well what the term “non-purified semolina” means.

Lanes 65-85: the methods are in general poorly described. Give more details and explain the principles. The methods used are not very common and therefore more information needs to be showed to understand the experiments done. Deliquescence of gluten is not a common trait used to analyze gluten for pasta-making. Gluten quality in durum is usually analyzed by Glutomatic, SDS-sedimentation, etc. Authors should try to use more common methods.

Lane 91: why wholegrain pasta was produced? Why not “normal” pasta done with semolina? Pasta industry uses mostly (>95%) refined semolina. The design of the study is poorly described.

Lane 94: flour or semolina? If flour was used, why?

Lanes 132-138: this is more for Materials and Methods and Discussion sections than in the Results section.

Lanes 137-138: again, please, do not over-generalize the results. In common wheat, there are hard and soft grain cultivars. The same for spelt wheat.

Lanes 167-174: this is pure discussion.

Lanes 197-204: again, this is discussion. And not very relevant. The same case for protein content data.

Lanes 218-240: most of this text should be in either the Materials and Methods or Discussion sections.

Lane 272: there is not discussion section. So, I realize now that probably results and discussion were merged. If this is the case, I recommend to split both sections and to do a proper discussion of the results, which is poor and incomplete.

Lanes 275-276: you should not compare with common wheat, which is rarely used to make pasta. You should compare with durum.

Lanes 282-284: I think the authors were too ambitious when taken conclusion of their data. Emmer was very bad for yellow color and for the organoleptic was even worse than spelt. Are you sure these emmer genotypes could be an alternative for durum for pasta-making?

Author Response

Thanks a lot to the Reviewer for a detailed analysis of our manuscript and constructive comments. All comments were taken into account when proofreading work. We've thoroughly corrected according to Reviewer's suggestion:

Lanes 29-32: change this paragraph. It seems that the authors are saying that hulled wheats have received lot of attention for pasta production in the last year. That is not true. Hulled wheats have received lot of attention to make traditional products, but not real interest by the pasta-making industry.

Lanes 29-32 We specified that it is about traditional products.

Lanes 33-34: please be more objective. SOME hulled wheat genotypes have higher concentration of some components related with nutritional quality but not all. There are some naked wheats (landraces and modern wheat) with better nutritional quality than some hulled wheats. Please, do not over-generalize the results of some studies. Hulled wheats are good, but they are not better for everything, including grain quality.

Lanes 33-34 Although we have not heard such reports, we have softened our statement.  Many literature data explanted a high nutritional value, including higher protein content and gluten yield as well as higher content of macro-, micronutrients and vitamins in hulled wheats . Our present study also these confirm.

Lanes 46-47: I cannot understand this well. Please, write it again.

We wrote it again: Some cultivars of common wheat are of a high technological value and are used not only as a bakery raw material, but also for the pasta production in some parts of Europe and the world.

Lane 50: is Floradur a “local” variety? I think this cultivar was developed in Austria.

Lane 50: We corrected this: The aim of the present study was to determine the usefulness of two species of hulled wheats, emmer and spelt, for pasta industry, as compared to common wheat and two cultivars of durum wheat grown in Poland under temperate conditions, with the indication that 'SMH87' is a local cultivar.

Lane 50: are local cultivars

not 'are' should be 'is' because only SMH87 is local cultivar . This was due to an unfortunate translation. We wrote the sentence again.

Lanes 29-50: the introduction is probably too short. The background given is not enough to understand why it is interesting to explore the pasta-making properties of hulled wheats. It is important to explain better why these lower yield and more expensive crops could be interesting to be used instead of durum wheat (lower price and better processing and end-use quality) by the pasta-making industry.

Lanes 29-50: We have greatly expanded the introduction and explained why are looking for  alternative and attractive for consumers pasta raw materials. It is diferrent in countriest when durum cultivation isn't simple. Thank you for pointing out the problem in this matter, because we will explain it in our article. Other reviewers did not have such questions, these issues were clear, but we may have different readers. And we want to be understood by everyone.

Lane 62: although the field trial is described in more detail in a previous paper, give some basic information of the trial in the current paper. Was it replicated? Type of design. What type of field management was applied?

Lane 62: We have added some information about the conditions of the field experiment, i.e. The experiment was continued for three consecutive years (2015-2017) in the locality of Hopkie (50°30'28" N 23°39'40" E; 221 m a.s.l., Lubelskie Province, Poland) on rendzina soil. The experiment was set up using a random block design, with three replicates, on plots with area of 0.17 ha. The conducted soil tillage was typical of a conventional tillage system and the fertilization and cultivation of the plantations were used as modern agriculture and managed in accordance with Good Agriculture Practice.

Lanes 78-80: this is not clear. I do not understand well what type of sample was used for the analysis showed below. Particularly, I do not understand well what the term “non-purified semolina” means.

Lanes 78-80: It should be whole-grain semolina. This was due to an inaccurate translation, we have corrected it everywhere in the text. We wrote the sentence again as follows: "From the wholegrain samples semolina were separated reaching - for all of tested wheat species -  the constant yield of 50% in relation to the grain."

Lanes 65-85: the methods are in general poorly described. Give more details and explain the principles. The methods used are not very common and therefore more information needs to be showed to understand the experiments done. Deliquescence of gluten is not a common trait used to analyze gluten for pasta-making. Gluten quality in durum is usually analyzed by Glutomatic, SDS-sedimentation, etc. Authors should try to use more common methods.

Lanes 65-85: Of course, we have developed a description of the individual methods. Unfortunately, we did not have the option of gluten quality analyzed by Glutomatic and SDS-sedymentation. In the Polish reality of pasta production, precisely these methods are used (indicated by the Polish Standards). Manual analysis allows experienced lab technicians to easily assess the quality of gluten and is widely used, among others deliquescence of gluten. We used the methods that we know well and that are widely used in the industrial evaluation of semolina quality.

Lane 91: why wholegrain pasta was produced? Why not “normal” pasta done with semolina? Pasta industry uses mostly (>95%) refined semolina. The design of the study is poorly described.

Line 91: The popularity of wholegrain pasta is increasing due to its nutritional and pro-health value, and growing consumer awareness . Therefore, we wanted to check whether pasta prepared from wholegrain semolina would have the right organoleptic parameters, because this raw material could turn out to be more technologically demanding.

Lane 94: flour or semolina? If flour was used, why?

Lane 94: It was semolina used of course. This was due to an unfortunate translation

Lanes 132-138: this is more for Materials and Methods and Discussion sections than in the Results section.

Lanes 132-138: We transferred some information to Materials and Methods. The discussion remained linked to the results to accordance with the guidelines of other Reviewers

Lanes 137-138: again, please, do not over-generalize the results. In common wheat, there are hard and soft grain cultivars. The same for spelt wheat.

These are colloquial terms, of course, we have removed the word "soft".

Lanes 167-174: this is pure discussion.

We added more english-language references and we have new-edited the discussion.

Lanes 197-204: again, this is discussion. And not very relevant. The same case for protein content data.

We added more english-language references and we have new-edited the discussion.

Lanes 218-240: most of this text should be in either the Materials and Methods or Discussion sections.

We wanted to outline the background for discussing the results in the right place. Following the suggestion, lines 218-223 have been moved to the Methodology

Lane 272: there is not discussion section. So, I realize now that probably results and discussion were merged. If this is the case, I recommend to split both sections and to do a proper discussion of the results, which is poor and incomplete.

It was indeed the merget "Results and Discussion". We completed the extensive discussion, but kept it linked to the results and gave a new title to the chapter "Result and Discussion" as suggested by the other Reviewer and our preferences.. We hope that the discussion in its new form is sufficient.

Lanes 275-276: you should not compare with common wheat, which is rarely used to make pasta. You should compare with durum.

The comparison to common wheat was actually unfortunate at the beginning of the conclusions. However, we wanted to emphasize that many parameters of hulled wheat are better than those common wheat, which is used to the pasta production in some world region.

Lanes 282-284: I think the authors were too ambitious when taken conclusion of their data. Emmer was very bad for yellow color and for the organoleptic was even worse than spelt. Are you sure these emmer genotypes could be an alternative for durum for pasta-making?

Yes, we are convinced that hulled wheat can be a good raw material for pasta production, at least in areas with difficult access to durum. While we were indeed too enthusiastic about our conclusions, we improved it. Organoleptic characteristics are not the most important thing for informed consumers if it is justified by the nutritional value of the products. The obtained results are promising and allow to plan a partial addition of hulled wheat semolina to traditional semolina as an increase in the range, which is in our further research plans.

We would like to thank the Reviewer for any constructive comments that will contribute to the improvement of our publication. We hope we corrected everything that was needed. However, if you feel there is something else left, please let us know. We are ready to improved our manuscript to make it good and published in Foods journal.

Reviewer 2 Report

In the article is recommended as the raw material between durum wheat also hulled emmer wheat that achieved with durum wheat positive comparable properties and possibilities of use in Central Europe.

To the article, I have next comments and recommendations:

  • In the article should be presented in all used equipment producer, town and country. It is used in some cases, but in others they are missing, like Brabender hardness tester, Quadrumat Junior mill, etc. Check thoroughly through the whole manuscript and complete the missing data.
  • Units: 200 g; 14 g, etc. be careful about the space.
  • In the References: you should use either the full names of the cited journals or their abbreviations, but not the mixture, you should this unify according the rules of the journal Foods. Do not use a mixture.
  • In Latin names use italics, also in References, e.g., Triticum monococcum, Triticum dicoccum,

Author Response

Thanks a lot to the Reviewer for an analysis of our manuscript and comments. All comments were taken into account when proofreading work. We've corrected:

We presented in all used equipment producer, town and country i.e.

  • line 69 Brabender hardness tester (Nossen, Germany)
  • line 76 Quadrumat Junior mill (Duisburg, Germany)
  • line 84 SD Matic (Chopin, Villeneuve-la-Garenne, France),
  • line 87 CR-410 Chroma Meter (Konica -Minolta, Tokyo, Japan)

We have carefully checked and corrected all spaces near units and  in all text.

We have introduced abbreviations for all journals in the References and used italic for latin name.

Reviewer 3 Report

The paper investigates hulled grains as alternative raw material for industrial pasta production.

The article is well written and the subject of the study is worth of investigation, nevertheless further information should be given in order to improve the scientific soundness of the paper.

In details:

L50: 'which are local cultivars'

L57: Check the correct scientific name of the wheat:

Durum:Triticum turgidum ssp. durum (Desf.); Emmer: Triticum turgidum ssp. dicoccum Schrank; common: Triticum aestivum ssp.aestivum L.; spelt : Triticum aestivum ssp. spelta L.

L70: What about Single Kernel Characterization System (SKCS)? It is the most direct system to measure kelnel texture.

L78: what do you mean by:' non-purified semolina' ?

L93: 'at room temperature'

L107: al dente, should be written in italics.

L109: What PZZ Lubella is?

L115: Could you explain what the honest significance difference mean?

L117: You should add Results and Discusion

L128: same comment of line 78

L150-151: It is not just a matter of starch damage

L161:What does p.p. stand for?

L204: Could you discuss the falling number values? What did you infer from these observations?

L291: There are too many references by L. Rachon (self citations)

In addition, Could you substitute some references in Polish language with some other in English?

Author Response

Thanks a lot to the Reviewer for an analysis of our manuscript and comments. All comments were taken into account when proofreading work. We've exactly corrected:

L50: 'which are local cultivars'

      We corrected this: The aim of the present study was to determine the usefulness of two species of hulled wheats, emmer and spelt, for pasta industry, as compared to common wheat and two cultivars of durum wheat grown in Poland under temperate conditions, with the indication that 'SMH87' is a local cultivar. Only SMH87 is local cultivar. It was unfortunately translation

L57: Check the correct scientific name of the wheat:

Durum:Triticum turgidum ssp. durum (Desf.); Emmer: Triticum turgidum ssp. dicoccum Schrank; common: Triticum aestivum ssp.aestivum L.; spelt : Triticum aestivum ssp. spelta L.

            L 57 we corrected Latina name of wheat according to Index Kewensis (based on https://www.gbif.org)  previously we based on https://theplantlist.org

L70: What about Single Kernel Characterization System (SKCS)? It is the most direct system to measure kelnel texture.

            L 70 unfortunately, we were unable to perform such analyzes

L78: what do you mean by:' non-purified semolina' ?        

            L 78: It should be whole-grain semolina. This was due to an inaccurate translation, we have corrected it everywhere in the text. We wrote the sentence again as follows: " From the wholegrain samples semolina were separated reaching - for all of tested wheat species -  the constant yield of 50% in relation to the grain."

L93: 'at room temperature'

            L 93 we changed 'of' to 'at' - thank you for this correction

L107: al dente, should be written in italics.

            L 107 'al dente' we corrected to italics

L109: What PZZ Lubella is?

            L 109 PZZ Lubella is polish lider of pasta producer , but we removed it from text  (https://www.lubella.pl/pl/produkty/makarony)

L115: Could you explain what the honest significance difference mean?

            L 115 Tuckey's test HSD used to determine a statistically significant difference. This is a commonly used test for such analyzes

L117: You should add Results and Discusion

            L 117 we added Result and Discussion

L128: same comment of line 78

            L 128: we corrected non-purified semolina to whole-grain semolina. It was due to an inaccurate translation

L150-151: It is not just a matter of starch damage

L150-151: Of course, we've added a some more information to this.

L161:What does p.p. stand for?

p.p. means 'percentage points'

L204: Could you discuss the falling number values? What did you infer from these observations?

We have added a discussion about the significance of the obtained results for the falling number

L291: There are too many references by L. Rachon (self citations)

In addition, Could you substitute some references in Polish language with some other in English?

We reduced the number of cited article by L. Rachoń, and substituted some references in Polish language with some other in English

We would like to thank the Reviewer for all constructive comments that will contribute to the improvement of our publication.

Round 2

Reviewer 1 Report

-

Author Response

-

Reviewer 3 Report

I believe the authors have satisfied all my review requests

Author Response

Thank you for your comment.